# Identification of an Attenuated Substrain of *Francisella tularensis* SCHU S4 by Phenotypic and Genotypic Analyses

**DOI:** 10.3390/pathogens10060638

**Published:** 2021-05-22

**Authors:** Julie A. Lovchik, Douglas S. Reed, Julie A. Hutt, Fangfang Xia, Rick L. Stevens, Thero Modise, Eileen M. Barry, Terry H. Wu

**Affiliations:** 1Center for Infectious Disease & Immunity and Department of Internal Medicine, University of New Mexico Health Sciences Center, Albuquerque, NM 87131, USA; lovchikja@gmail.com; 2Center for Vaccine Research, University of Pittsburgh, Pittsburgh, PA 15261, USA; dsreed@pitt.edu; 3Lovelace Respiratory Research Institute, Albuquerque, NM 87108, USA; jhutt@gfpath.com; 4Data Science and Learning, Argonne National Laboratory, Argonne, IL 60439, USA; fangfang@anl.gov; 5Computing, Environment and Life Sciences, Argonne National Laboratory, Argonne, IL 60439, USA; stevens@anl.gov; 6Department of Computer Science, The University of Chicago, Chicago, IL 60637, USA; 7Department of Biomedical Sciences, Faculty of Medicine, University of Botswana, Gaborone, Botswana; modiseth@ub.ac.bw; 8Center for Vaccine Development, University of Maryland School of Medicine, Baltimore, MD 21201, USA; embarry@som.umaryland.edu

**Keywords:** *Francisella tularensis*, SCHU S4, attenuation, virulence, substrain, rats, rabbits

## Abstract

Pneumonic tularemia is a highly debilitating and potentially fatal disease caused by inhalation of *Francisella tularensis.* Most of our current understanding of its pathogenesis is based on the highly virulent *F. tularensis* subsp. *tularensis* strain SCHU S4. However, multiple sources of SCHU S4 have been maintained and propagated independently over the years, potentially generating genetic variants with altered virulence. In this study, the virulence of four SCHU S4 stocks (NR-10492, NR-28534, NR-643 from BEI Resources and FTS-635 from Battelle Memorial Institute) along with another virulent subsp. *tularensis* strain, MA00-2987, were assessed in parallel. In the Fischer 344 rat model of pneumonic tularemia, NR-643 and FTS-635 were found to be highly attenuated compared to NR-10492, NR-28534, and MA00-2987. In the NZW rabbit model of pneumonic tularemia, NR-643 caused morbidity but not mortality even at a dose equivalent to 500x the LD_50_ for NR-10492. Genetic analyses revealed that NR-10492 and NR-28534 were identical to each other, and nearly identical to the reference SCHU S4 sequence. NR-643 and FTS-635 were identical to each other but were found to have nine regions of difference in the genomic sequence when compared to the published reference SCHU S4 sequence. Given the genetic differences and decreased virulence, NR-643/FTS-635 should be clearly designated as a separate SCHU S4 substrain and no longer utilized in efficacy studies to evaluate potential vaccines and therapeutics against tularemia.

## 1. Introduction

Tularemia is caused by the highly infectious Gram-negative bacterium, *Francisella tularensis* [1]. The two clinically important subspecies of *F. tularensis* are subspecies *tularensis* (type A) and subspecies *holarctica* (type B). While both subspecies are highly infectious, type A strains are the most virulent and cause the most serious form of the disease when inhaled as an aerosol [1]. Without prompt and appropriate treatment, the disease can progress very rapidly and cause sepsis, shock, and death. It is estimated that 30 to 60% of patients who developed pleuropulmonary tularemia died from this disease without effective therapy [2]. Because of its relatively high respiratory infectivity and virulence, *F. tularensis* is considered a potential bioweapon, and was designated by the United States Centers for Disease Control and Prevention (CDC) and National Institute of Allergy and Infectious Disease (NIAID) as a Tier 1, Category A select agent. Currently, there is no licensed tularemia vaccine, and frontline antibiotics may become ineffective if the bacteria develop natural or engineered resistance. Therefore, the development of novel vaccines and therapeutics is a high priority in the national biodefense program. 

The natural incidence of tularemia in humans is very low; as such, evaluation of new potential vaccines and therapeutics against tularemia depends on well-characterized animal models that mimic the human disease. The highly virulent SCHU S4 strain has long been utilized as the prototypic type A strain of *F. tularensis* for use in such models [3]. Since its original isolation in 1951 [4], the SCHU S4 strain has been used extensively to study the pathogenesis of pneumonic tularemia in both humans and experimental animals [5,6,7], as well as to evaluate the efficacy of potential medical countermeasures [5,8,9,10,11,12,13].

SCHU S4 has been consistently virulent in animal models ranging from mice to non-human primates (NHP) over many years [14,15,16,17,18,19]. However, in a study by Molins, et al. utilizing a mouse tularemia model, SCHU S4 was reported to be less virulent than a more recent type A clinical isolate from Massachusetts, MA00-2987 [20]. Multiple sources of SCHU S4 have been maintained and continually passaged for many years, so it was not clear whether this apparent reduction in virulence was specific to the stock used, a result of the choice of animal model, culture method, route of inoculation, or reflective of a genuine difference in virulence between the two strains [4]. Thus, the goal of the current study was to evaluate in parallel the genotype and phenotype of several SCHU S4 stocks obtained from BEI Resources (catalog no. NR-28534, NR-10492, and NR-643) and Battelle Memorial Institute (FTS-635). For these studies, MA00-2987 (BEI Resources catalog no. NR-645) was used as a comparator.

## 2. Results

### 2.1. SCHU S4 Stocks Segregate into Two Distinct Virulence Phenotypes in F344 Rats

In this study, the virulence of four separate SCHU S4 stocks was evaluated in parallel with MA00-2987 in the Fischer 344 (F344) rat model of pneumonic tularemia. Additional studies were also conducted comparing two of the four SCHU S4 stocks in the NZW rabbit model of pneumonic tularemia. Three SCHU S4 stocks were obtained from BEI Resources (catalog no. NR-28534, NR-10492, and NR-643): NR-28534 was a master cell bank deposited by the U.S. Department of Defense Joint Vaccine Acquisition Program (JVAP); NR-10492 was a submaster cell bank prepared by direct dilution from NR-28534; NR-643, which was the stock utilized in the mouse study by Molins, et al. [20], was deposited by the CDC. The fourth stock (FTS-635) was obtained from Battelle Memorial Institute and it was derived from colonies originally provided by Dr. John Gunn at Ohio State University. These four SCHU S4 stocks are hereinafter referred to by the catalog numbers used at BEI Resources or Battelle Memorial Institute (Table 1). MA00-2987, which was isolated from a fatal human case of pulmonary tularemia in the 2000 tularemia outbreak on Martha’s Vineyard [21], was also obtained from BEI Resources (catalog no. NR-645).

The virulence of the four SCHU S4 stocks and MA00-2987 was first compared in F344 rats using a target lung deposition approximately 100-times the calculated 50% lethal dose (LD_50_) for aerosolized NR-28534 in the F344 rat model [15]. As shown in Figure 1, all rats exposed to aerosolized NR-28534 and NR-10492 died between 4 and 8 days after challenge, and a similar pattern of survival was observed in rats exposed to the MA00-2987. In contrast, all of the rats exposed to NR-643 or FTS-635 survived to study termination, indicating that they were less virulent compared to NR-28534, NR-10492, or MA00-2987.

We next compared the natural history of disease in rats infected with NR-10492 and NR-643. Consistent with the previous findings for NR-10492, a dose of 212 CFU was lethal for all exposed animals (Figure 2A). Infected rats rapidly lost weight (Figure 2B) and developed additional clinical signs as the disease worsened, beginning with a ruffled coat on Day 4 and progressing to extremely labored breathing, weakness and/or ataxia, and hunched posture when rats became moribund. Death occurred between 5 and 7 days after exposure. In contrast, 9 of 10 rats exposed to 486 CFU of NR-643 survived infection (Figure 2A). The surviving rats all developed mild clinical signs of illness and lost weight but began to recover approximately 2 weeks after challenge (Figure 2B). 

Both NR-10492 and NR-643 were able to replicate in the lungs and disseminate systemically to the liver and spleen. However, fewer bacteria were recovered from the lungs, liver, and spleen of the NR-643-infected rats on day 3 than the NR-10492-infected rats (Figure 2C). When NR-10492-infected rats were either found dead or euthanized having met the euthanasia criteria, the lungs contained 2 × 10^8^ to 3 × 10^10^ CFU/g of tissue and the spleens contained 4 × 10^8^ to 2 × 10^10^ CFU/g of tissue, suggesting that NR-10492 had continued to proliferate aggressively in the infected animals after Day 3. The results were consistent with historical data from rats that succumbed after exposure to aerosolized NR-28534. In contrast, the NR-643 infected rats constrained the infection and, by study termination on Day 20, only small numbers of bacteria (60 to 4085 CFU/g) were recovered from the lungs of all nine surviving rats. Five of those rats had completely cleared the infection from the spleen.

The microscopic pathology findings in the rats infected with 212 CFU of NR-10492 or 486 CFU of NR-643 were characteristic of primary pneumonic tularemia with hematogenous dissemination to the liver and spleen [15]. On day 3, neutrophilic and histiocytic inflammation was present in the lungs (within alveoli, bronchioles, and the peribronchovascular region), lymph nodes (primarily cortex), livers (multiple randomly located foci in the hepatic parenchyma), and spleens (multiple randomly located foci within the splenic red pulp) of all animals. The severity of the lung inflammation in the rats infected with NR-643 was greater than in the rats infected with NR-10492 (Figure 3). In contrast, the severity of the inflammation in the liver and spleen for rats infected with NR-643 was similar (liver) or less (spleen) than for animals infected with NR-10492, consistent with reduced dissemination for NR-643. At the time of death or moribund euthanasia between days 5 and 7, the inflammation in the lungs, lymph nodes, liver, and spleen of all NR-10492 infected rats and the one NR-643-infected rat that succumbed to infection was more severe than on day 3, and consisted of neutrophilic and histiocytic cellular infiltrates with fibrin exudation and necrosis of inflammatory cells and endogenous tissue elements. The livers of many rats also exhibited sinusoidal microthrombi and necrotic cell debris. In addition, there were decreased lymphocytes in the splenic white pulp of all rats. When the nine surviving NR-643-infected rats were euthanized at study termination on day 20, microscopic pathology findings were still detected in the lungs, lymph nodes, livers, and spleens, but were present at reduced severity levels than in the animals that succumbed and were consistent with a chronic/resolving inflammatory process. 

In an additional experiment, groups of F344 rats were infected with increasing doses of NR-643 to measure the level of attenuation compared to NR-10492. The results showed that mortality was dose-dependent, and all rats succumbed to NR-643 infection at a dose of 3255 CFU or higher between day 4 and 8 post-infection (Figure 4A). The clinical presentation and disease progression were indistinguishable between rats challenged with lethal doses of NR-643 or NR-10492 (Figure 4B). Thus, NR-643 appeared to be significantly attenuated compared to NR-10492, but not completely avirulent in F344 rats.

### 2.2. NR-643 Is Significantly Attenuated Compared to NR-10492 in NZW Rabbits

Naïve NZW rabbits succumbed within 4–7 days after aerosol infection with NR-10492, at doses ranging from as low as 83 to 53,000 CFU (n = 53) [19]. The median dose for exposure to NR-10492 was 4203 CFU; 13 rabbits were exposed to a dose less than 1000 CFU. Similar to what was seen with F344 rats, NZW rabbits infected with aerosol doses of NR-643 (n = 5) ranging from 54 to 11,010 CFU survived until study termination at day 12 post-challenge (Figure 5A). The distribution of NR-643 doses in rabbits was as follows: 54, 256, 4243, 5523, 11,010 CFU. NR-643-infected rabbits did develop fever in response to infection, but fever was transient with temperatures returning to normal levels by study termination, whereas fever persisted in NR-10492-infected rabbits until animals succumbed to infection (Figure 5B). NR-643-infected rabbits also had only a moderate weight loss when challenged, compared to severe, progressive weight loss in rabbits infected with NR-10492 (Figure 5C). Organ bacterial burdens at study termination were significantly reduced and or undetectable in NR-643-infected rabbits as compared to the high terminal bacterial burdens seen in rabbits that succumbed to similar or lower challenge doses of NR-10492 (Figure 5D).

### 2.3. Genomic DNA Sequencing Identifies NR-643 and FTS-635 as a Substrain of SCHU S4

To determine whether the phenotypic differences observed among the SCHU S4 stocks reflected genotypic differences and to determine the relationship between the four SCHUS4 stocks and MA00-2987, genomic DNA sequencing was performed by the Broad Institute. The results were compared against the published SCHU S4 reference genomic DNA sequence (GenBank accession number AJ749949.2; Table 2) using read mapping as well as reference-assisted and de novo assemblies. Because each of these variant discovery methods is associated with known biases and artifacts [22], the region of difference (RD) between the SCHU S4 stocks and the reference sequence were identified from the consensus of the four analyses: (1) Alignment between reference-assisted contigs and the reference, (2) alignment between de novo contigs and the reference, (3) variant calling by mapping reads to the reference, and (4) de novo assembly of reads found in regions overlapping repeats. The results from aligning the four assemblies revealed that NR-28534 and NR-10492 were identical to each other but differed from the reference sequence by the presence of an inversion of a region 25,683 base pairs (bp) in length (Table 3). This inversion is not unusual since inversions have been observed in other virulent type A1 strains such as NE061598 [23,24] and FSC033. In addition, NR-28534 and NR-10492 contained a C → T intergenic single nucleotide polymorphism (SNP) located between FTT1698c and FTTr08. There were no other statistically significant SNPs found between NR-28534, NR-10492, and the reference sequence. Comparison between MA00-2987 and the reference sequence using the dnadiff from MUMmer [25] to align the sequences showed that MA00-2987 was 99.86% identical to the reference sequence at the DNA level, and 94% of its proteins had homologs in the reference strain. NR-643 and FTS-635 were identical to each other but shared nine putative RDs with respect to the reference genome (Table 3). These included six SNPs (one intergenic and five nonsynonymous), one single-base insertion, one single-base deletion that causes a frameshift, and one 11 bp intergenic deletion. Mapping of read data for NR-643 and FTS-635 on the reference sequence indicated the presence of variants at the 11 bp deletion site; specifically, half of all reads that covered the 11 bp deletion site in both NR-643 and FTS-635 had a deletion while the other half did not. These results suggested that both NR-643 and FTS-635 were composed of a mixed population. The effect of the frameshift at reference location 427432 is unknown but did not affect the number of predicted ORFs. There appeared to be no difference between NR-643 and FTS-635 that could not be attributed to assembly errors. Altogether, these results show that NR-643 and FTS-635 represent a substrain of SCHU S4 that is separate and significantly different from NR-28534 and NR-10492.

## 3. Discussion

As novel tularemia vaccines and therapeutics are developed, it is critical that their efficacy be evaluated in an appropriate animal model using a virulent strain of *F. tularensis*, typically from subsp. *tularensis* (Type A). SCHU S4 has historically been the prototypic type A strain for use in such efficacy studies and has been consistently virulent in animal models ranging from mice to NHP. However, multiple sources of SCHU S4 isolates have been maintained over the years, and there is always a risk of genetic alteration due to continued passage and/or maintenance in multiple labs [4]. Two such spontaneously derived mutants of SCHU S4 have been identified and are referred to as SCHU S4 P0 and FSC043 [26,27]. Both substrains were shown to be significantly attenuated in murine challenge models, and genomic sequencing showed that both variants contained multiple mutations [26,28,29]. In the current study, the characterization of four available SCHU S4 stocks has now revealed the existence of another highly attenuated substrain of SCHU S4 containing multiple genetic differences compared to the SCHU S4 reference strain.

NR-28534, NR-10492, NR-643, and FTS-635 are currently all considered to be *F. tularensis* SCHU S4. The results of the current study confirmed that both the master (NR-28534) and submaster (NR-10492) cell banks of SCHU S4 deposited by NIAID with BEI Resources are genetically identical and only differ from the published reference SCHU S4 sequence by an 25,683 bp inversion and an intergenic SNP. They exhibited the expected phenotype of SCHU S4 based on historical data and were highly lethal in F344 rats, similar to the *F. tularensis* type A strain, M00-2987. In addition, NR-10492 has also been shown to be highly lethal in naive NZW rabbits (this study and previously published studies [19,30,31,32,33] and the cynomolgus macaque (*Macaca fascicularis*) [14,18]. In contrast, NR-643 and FTS-635 were highly attenuated in F344 rats, requiring greater than 20-fold higher challenge dose to cause 100% lethality as compared to NR-28534 and NR-10492. Similarly, NR-643 was attenuated in NZW rabbits with no mortality and pathogenesis confined to the lungs. The finding that NR-643 and FTS-635 were genetically identical explained the similar phenotype exhibited in the F344 rat model. These findings also explain the results in a previous mouse study which utilized NR-643, and therefore concluded that SCHU S4 was less virulent than other *F. tularensis* type A strains, such as the MA00-2987 strain [20]. Thus, NR-643/ FTS-635 has been confirmed to be less virulent than the prototypic *F. tularensis* SCHU S4 and/or another type A strain in three separate labs and three separate animal models of tularemia. 

The genotypic differences between NR-643/FTS-635 and the SCHU S4 reference strain included a SNP in FTT_0807 resulting in an amino acid change in CapA, which has a predicted role in polysaccharide assembly [34], a frameshift mutation in *glgC*, which encodes a putative glucose-1-phosphate adenylyltransferase, and amino acid changes in the *fabF* and *fabH* genes, which encode two separate enzymes in the bacterial type II fatty acid synthesis (FAS-II) pathway. However, it is unclear at this time which one, or combination of the nine identified genetic mutations, was responsible for the attenuated virulence observed in the NR-643/FTS-635 substrain. Unlike the FSC043 and SCHU S4 P0 SCHU S4 mutant strains, which both lacked functional PdpC and were essentially avirulent in mice [26,27,28], no mutations were found within the critical virulence genes of the *Francisella* pathogenicity island (FPI) in NR-643/FTS-635. Although the virulence of NR-643/FTS-635 was reduced, it was not completely abolished. Aerosol exposure to ≥3255 CFU of NR-643 in F344 rats was lethal and induced a similar pattern of clinical signs, weight loss, and kinetics of disease progression as seen following infection with a lethal dose of NR-10492. NR-643 was also able to proliferate in F344 rats and spread from the lung to other organs, although on day 3, organ bacterial burdens in NR-643 infected rats were significantly lower than organ burdens in rats infected with NR-10492 at a comparable challenge dose. Furthermore, most of the surviving NR-643 infected rats had completely cleared the organism by study termination on Day 20 post-infection. Bacterial burdens in the spleens and livers of NZW rabbits infected with NR-643 were low to undetectable at study termination and gross pathology suggested little or no dissemination to other organs had occurred. These data suggest that the genetic differences in NR-643/FTS-635 may make it more susceptible to host defenses compared to NR-10492 and less able to escape the lung. Additional studies will be required to dissect the actual genetic mutation(s) responsible for the decreased virulence of NR-643/FTS-635. 

In summary, the NR-643/FTS-635 strain is significantly attenuated compared to the NR-28534/NR-10492 SCHU S4 strain or the MA00-2987 strain of *F. tularensis*, with multiple genetic differences compared to the SCHU S4 reference sequences. These results were confirmed in two laboratories and in different animal models. We would strongly suggest that NR-643/FTS-635 should no longer be designated as SCHU S4 to prevent confusion and use in future efficacy studies. These results also emphasize the need for reliable, well-characterized SCHU S4 challenge strains that are produced and passaged under defined conditions, as well as periodically sequenced in order to maintain genetic and phenotypic homogeneity for use in pivotal efficacy studies to support licensure of novel vaccines and therapeutics.

## 4. Materials and Methods

### 4.1. Animals

#### 4.1.1. Rats 

Female F344 rats between 2 and 3 months of age were purchased from Envigo (Indianapolis, IN, USA). The animals were housed in groups of 6 or less in individually ventilated cages (Techniplast, Milan, Italy) in an ABLS-2 laboratory before challenge and in a CDC-certified select agent ABSL-3 laboratory after challenge. The cages were maintained at a negative pressure relative to the facility. Fresh air was HEPA filtered going into the cages, and exhaust air was HEPA filtered and ventilated out of the facility. The cages were lined with Tek-Fresh pelleted, virgin paper fiber bedding (Envigo), and changed approximately every 2 weeks. The animals were fed ad libitum with Teklad irradiated rodent chow #2920x (Envigo) and given chlorine dioxide-treated filtered water. All animal housing rooms in the facility were under temperature and humidity control with a 12 h light on/off cycle. All work conducted with animals was approved by the University of New Mexico Health Sciences Center Institutional Animal Care and Use Committee. 

#### 4.1.2. Rabbits 

Female New Zealand White (NZW) Rabbits approximately 3 months in age were purchased from Robinson Services, Inc. (Mocksville, NC, USA). Rabbits were housed in the University of Pittsburgh RBL at ABSL-3 for the duration of the studies. All studies were performed under protocols approved by the University of Pittsburgh’s Institutional Animal Care and Use Committee. Rabbits were monitored at least once daily prior to infection and at least twice daily after infection until clinical signs resolved or the animal became moribund. Prior to study initiation, IPTT-300 temperature/ID chips (BioMedic Data Systems, Seaford, DE, USA) were implanted subcutaneously. 

### 4.2. Challenge Organisms

*Francisella tularensis* subsp. *tularensis* SCHU S4 Master Cell Bank, NR-28534, Submaster Cell Bank, NR-10492, and Strain SCHU S4 (FSC237), NR-643, were obtained through BEI Resources, NIAID, NIH (Manassas, VA, USA). *Francisella tularensis* subsp. *tularensis*, Strain MA00-2987, NR-645 was obtained through the NIH Biodefense and Emerging Infections Research Resources Repository, NIAID, NIH. The SCHU S4 substrain FTS-635 was obtained from Battelle Memorial Institute (West Jefferson, OH, USA). Working stocks were prepared from the materials provided by inoculating approximately 200 to 250 μL of scrapings into 25 mL Modified Cysteine Partial Hydrolysate (MCPH) in 125 mL disposable Erlenmeyer flasks with vented cap (Corning; Corning, NY, USA). The cultures were incubated at 37 °C with constant rotation at a rate of 200 rpm for 22 to 44.5 h. The cultures were supplemented with 20% glycerol and stored in aliquots at −80 °C. Approximately 24 h after freezing, a vial from each working stock was thawed to determine the concentration of viable organisms and colony morphology. For rabbit work at the University of Pittsburgh, NR-643 and NR-10492 were subcultured on cysteine heart agar for two days before inoculation in 25 mL of Brain Heart Infusion (BHI) broth to prepare working stocks which were frozen in BHI with 20% glycerol as previously described [30,35]. All work performed with *F. tularensis* strains at both facilities were carried out using appropriate Biosafety Level 3 standard operating procedures in a CDC-registered ABSL-3 select agent laboratory.

### 4.3. Preparation of Challenge Organism for Aerosolization

#### 4.3.1. Rats 

Four days before challenge, a vial from each working stock was thawed to room temperature, plated onto cysteine heart agar plates supplemented with rabbit blood, 100 unit/mL penicillin G, and 100 unit/mL polymyxin B (CHAB; Thermo Fisher Scientific Remel Products, Lenexa, KS, USA), and cultured for 3 days at 37 °C. On the day before challenge, colonies approximately 1–2 mm in diameter were suspended into 4.5 mL Chamberlain’s Chemically Defined Medium (CCDM) to produce a bacterial suspension with an optical density at 600 nm (OD_600_) of 0.1. Four hundred microliters of the resulting bacterial suspension was inoculated into 100 mL sterile CCDM in a sterile 500 mL disposable Erlenmeyer flask with vented cap (Corning) and cultured at 37 °C ± 2 °C with 200 rpm rotary aeration for 17–20 h. The overnight cultures were diluted with sterile CCDM to an OD_600_ of 0.1, which was expected to contain 1.0 ± 0.2 × 10^8^ CFU/mL. The resulting bacterial suspensions were further diluted with sterile Brain Heart Infusion Broth (BHIB; Teknova, Hollister, CA, USA) to the concentration (CFU/mL) required to achieve the target lung deposition based on historical correlation between generator concentration and lung deposition. The generator solutions were serially diluted and plated on CHAB plates to determine the actual pre-spray bacterial inoculum concentration.

#### 4.3.2. Rabbits

Preparation of challenge organism was as previously described [35]. Three days prior to challenge a frozen working stock vial was removed from the −80 °C freezer and a small aliquot was streaked on a Cysteine Heart Agar (CHA) plate and incubated for 2 days at 37 °C. The day before challenge, colonies were taken from the plate and inoculated into BHI broth to an OD_600_ of 0.1. 500 μL were inoculated into 24.5 mL of BHI and put into a 125 mL polycarbonate vented baffled flask and incubated in a shaker incubator for 18 h at 37 °C, 200 RPM. The OD600 of the culture was read and the broth culture was diluted to the nebulizer concentration needed to achieve the desired target dose in BHI based on historical aerosol data [30].

### 4.4. Aerosol Exposure

#### 4.4.1. Rats

Aerosolized organisms were generated using a Collison 3-jet nebulizer (BGI, Inc., Waltham, MA, USA) and presented to the study animals in a nose-only exposure chamber (In-Tox Products, Inc., Moriarty, NM, USA). The generator flow rate was maintained at 7.5 L/min and the impinger flow rate was maintained at 5 L/min. The animals were exposed to aerosols for 15 min, and the system was purged for 2 min before the animals were removed. Between exposure runs with different stocks, the aerosol lines and animal exposure chamber were decontaminated by running 70% ethanol through the system for 15 min followed by sterile water for 15 min, and finally, a 2 min purge with air alone. Decontamination of non-porous surfaces with 70% ethanol for 15 min has been shown to reduce the number of SCHU S4 by at least 7 log_10_ [36]. An aerosol sample was collected 10 min into each exposure run for 20 s. The aerosol sample was diluted 1:20 using an aerosol dilutor (TSI; Shoreview, MN, USA) and analyzed using an aerodynamic particle sizer spectrometer (TSI). A sample from the all glass impinger (Ace Glass, Inc., Vineland, NJ, USA) was plated after each run onto CHAB plates, and the resulting colony counts were used to calculate the presented dose. Representative animals from each exposure run were euthanized after exposure and the lung homogenates were plated to determine the lung deposition. The average lung deposition from all exposure runs in the study with the same substrain was used as the challenge dose for that substrain. Purity of aerosolized sample was assessed by colony morphology.

#### 4.4.2. Rabbits

Aerosols of Ft were conducted inside a class III biological safety cabinet (Baker Co., Sanford, ME, USA) located inside the RBL as previously described using the Biaera AeroMP exposure system (Biaera Technologies, Hagerstown, MD, USA) [30]. Rabbits were exposed for 10 min in a nose-only exposure chamber (CH Technologies, Westwood, NJ, USA) using a vertical discharge 3-jet Collison nebulizer controlled by the AeroMP. Plethysmography was performed during the aerosol to measure rabbit minute volume using Buxco XA software (Data Sciences Inc, St. Paul, MN, USA). Aerosol samples were collected in an all-glass impinger containing BHI and antifoam A. These samples were plated on CHA to determine bacterial aerosol concentration. Presented or inhaled dose was determined as described previously, as the product of the aerosol concentration, the minute volume of the rabbit, and the duration of the aerosol.

### 4.5. Observations and Measurements

#### 4.5.1. Rats

Clinical observations were performed twice-a-day, 6 to 8 h apart after challenge, and body weights were obtained once a day. Animals were given a clinical score based on symptoms ranging from bright, alert, and responsive to moribund. Animals meeting moribund criterion were euthanized by overdose of sodium pentobarbital. Mortality was recorded on the day the animal was euthanized or found dead in accordance with approved IACUC protocol. 

#### 4.5.2. Rabbits 

Body weight was recorded once in the morning and body temperature was recorded twice daily. Temperature was read using a DAS-7000 reader (BioMedic Data Systems). Observations were increased to three times daily when rabbits were sick. Rabbits that were determined to be moribund (any of the following clinical signs: Weight loss ≥20%, body temperature <34 °C, unresponsive to prodding, respiratory distress) were first anesthetized with isoflurane (2–5%) and then euthanized promptly by barbiturate overdose (100 mg/kg sodium pentobarbital given i.v. or i.c.). Rabbits infected with NR-643 that survived to day 12 were euthanized by barbiturate overdose. All rabbits were given a complete necropsy to collect blood and tissue samples for bacteriological and pathological evaluation.

### 4.6. Genome Sequencing

Whole genome sequencing for the two NIAID *F. tularensis* cell banks were carried out at the Broad Institute using the HiSeq 2000 System (Illumina; San Diego, CA, USA). The stocks were not passaged more than once prior to sequencing. The DNA for each stock was sequenced to ~1500× total coverage with 102-bp reads using a combination of 180-bp fragment libraries and 5-kb sheared jumping libraries. Detailed sequencing information can be found in the Sequence Read Archive: SRP030145 for NR-28534, SRP030144 for NR-10492, SRP030146 for NR-643, and SRP030143 for FTS-635.

### 4.7. Genome Assembly and Annotation

Reference-assisted and de novo methods were used to assemble the SCHU S4 genomes. The reference-assisted assembly was done at the Broad Institute using the ALLPATHS-LG assembler [37] with the “ASSISTED_PATCHING=2.1” parameter. The raw assemblies were then postprocessed with the Pilon assembly improvement tool and manually reviewed with GAEMR (http://www.broadinstitute.org/software/gaemr/, accessed on 15 April 2021). These reference-assisted assemblies have been deposited to GenBank and assigned assembly accessions (see Table 1). The de novo assembly was done using the Assembly Service in PATRIC [38] with the SPAdes protocol: SPAdes with BayesHammer as the error corrector and the “--careful” option to minimize the number of mismatches in the final contigs [39]. Following genome assembly, genome annotations were completed using the PATRIC Annotation Service [38]. The software MIRA version 5 [40] was used for de novo assembly of highly repetitive regions using the parameter switch “--hirep_best”. Finally, to validate MIRA results and resolve complex regions with a high number of tandem repeats in the substrain NR-28534, PacBio reads with SRA data accession number SRR1714340 (for SCHU S4 stock NR-28534) were used.

### 4.8. Identifying Regions of Differences among Substrains

Whole-genome alignment as well as read mapping were used to identify regions of difference (RD) between the SCHU S4 stocks and the reference sequence. An RD is considered solid when it is the consensus of four analyses: (1) Alignment between reference-assisted contigs and the reference, (2) alignment between de novo contigs with the reference, (3) variant calling by mapping reads to the reference, and reference (4) de novo assembly of reads found in regions overlapping repeats. Whole-genome alignment was done using Mugsy [40], and mapping-based variant calling was done using BWA-MEM (H. Li, unpublished data) and FreeBayes (E. Garrison and G. Marth, unpublished data). Alignments were also done using bowtie2 [41] and the output files processed with SAMtools [42] to identify variants. The visualization of mapped reads was done via UGENE [43] using bowtie2 as a default aligner. The SAMtools software was also used to extract reads for regions with higher copy numbers of variable number of tandem repeats (VNTRs). The reads were then reassembled using MIRA5 to identify more accurate counts of the VNTRS as well as the accurate long sequence of the VNTRs. The program BLAST+ [44] was also used to confirm the number of VNTRs. Briefly, a database of reads for each sequencing run was created and queried with the VTNR element using the program blastn. The search parameters were adjusted for short subject sequences. 

### 4.9. Statistics

Comparisons of survival curves between different groups were statistically evaluated by Kaplan-Meier and Log-rank (Mantel-Cox) analysis (GraphPad Prism, Version 9.0.1, San Diego, CA, USA). Comparisons of organ bacterial burdens following challenge with the NR-643 or NR-10492 SCHU S4 strain were analyzed by repeated measures two-way ANOVA with Bonferroni’s multiple comparison test.

## Figures and Tables

**Figure 1 pathogens-10-00638-f001:**
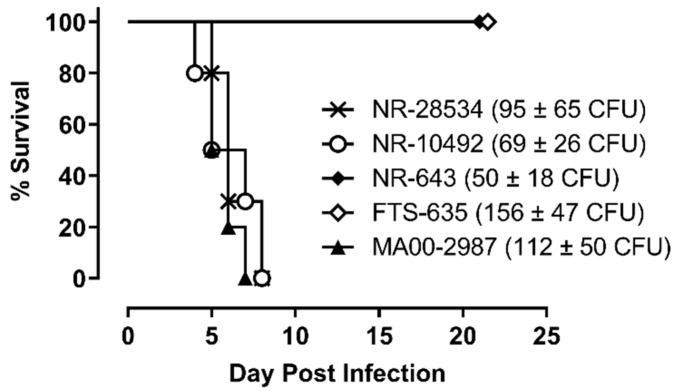
SCHU S4 with different levels of virulence. Female F344 rats (n = 10 per group) were exposed to aerosols containing *F. tularensis* strain SCHU S4 NR-643, FTS-635, NR-10492, or NR-28534 or *F. tularensis* strain MA00-2987 at the challenge doses indicated. Challenge doses reflect the average lung deposition from three representative rats challenged in parallel and euthanized after exposure. The calculated presented doses were approximately 4 to 7-fold higher than the average lung deposition. Survival was monitored for a period of 21 days.

**Figure 2 pathogens-10-00638-f002:**
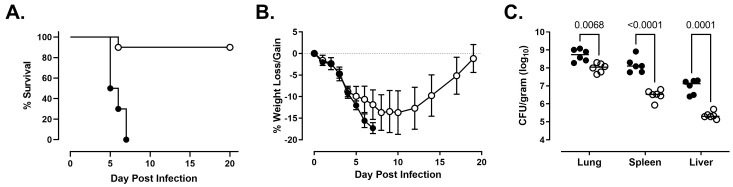
**NR-643 causes infection but is attenuated compared to NR-10492.** Female F344 rats (n = 10 group) were exposed to 212 ± 61 CFU of NR-10492 (●) or 486 ± 99 CFU of NR-643 (◯) by aerosol inhalation and monitored for survival (Panel **A**) and weight loss/gain (Panel **B**) for 20 days. Change in body weight was calculated relative to each animal’s weight on the day of exposure and presented as group average. Error bars shows SD. Six additional rats exposed to NR-10492 and NR-643 at the same time were euthanized 3 days after exposure to measure bacterial burden (Panel **C**). Significant differences in the bacterial burdens were observed between the rats infected with NR-10492 and NR-643 (*p* < 0.0001) and in the lungs, livers, and spleens.

**Figure 3 pathogens-10-00638-f003:**
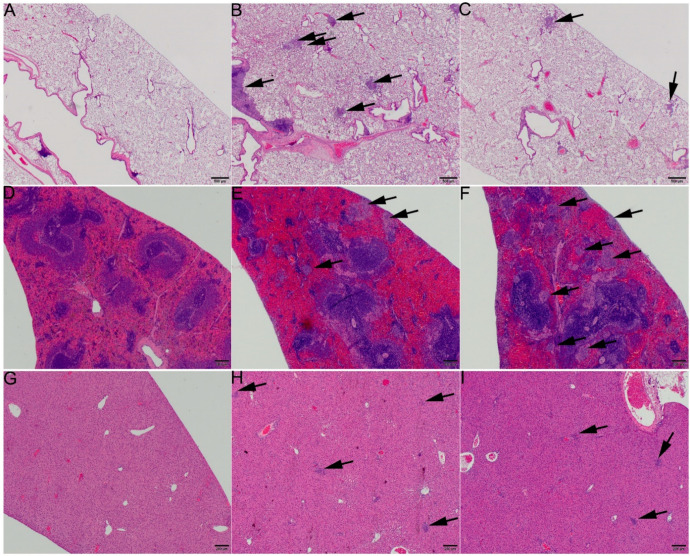
NR-643 caused more severe lung inflammation than NR-10492. Tissues from rats infected with 212 ± 61 CFU of NR-10492 or 486 ± 99 CFU of NR-643 were examined 3 days after aerosol exposure. (**A**,**D**,**G**) show lung, spleen, and liver from a naïve rat. (**B**,**E**,**H**) show lung, spleen, and liver from a rat challenged with NR-643 and (**C**,**F**,**I**) show lung, spleen, and liver from a rat challenged with NR-10492. Arrows point to foci of inflammation. Note the larger number of inflammatory foci in (**B**) compared to (**C**), and the similar (liver) or greater (spleen) number of inflammatory foci in (**F**,**I**) compared to (**E**) and (**H**). **H** and **E** stain. Scale bars = 500 microns.

**Figure 4 pathogens-10-00638-f004:**
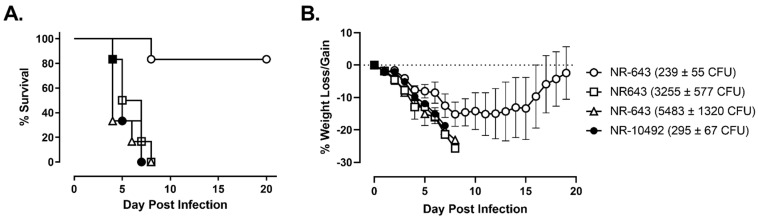
Lethal dose for NR-643 is 20-times higher than NR-10492. Female F344 rats (n = 6/group) were exposed to the indicated doses of NR-643 and NR-10492 by aerosol inhalation and monitored for survival (Panel **A**) and weight loss/gain (Panel **B**). Change in body weight was calculated relative to each animal’s weight on the day of exposure and presented as group average. Dotted line shows percent weight change on the day of exposure and error bars shows SD.

**Figure 5 pathogens-10-00638-f005:**
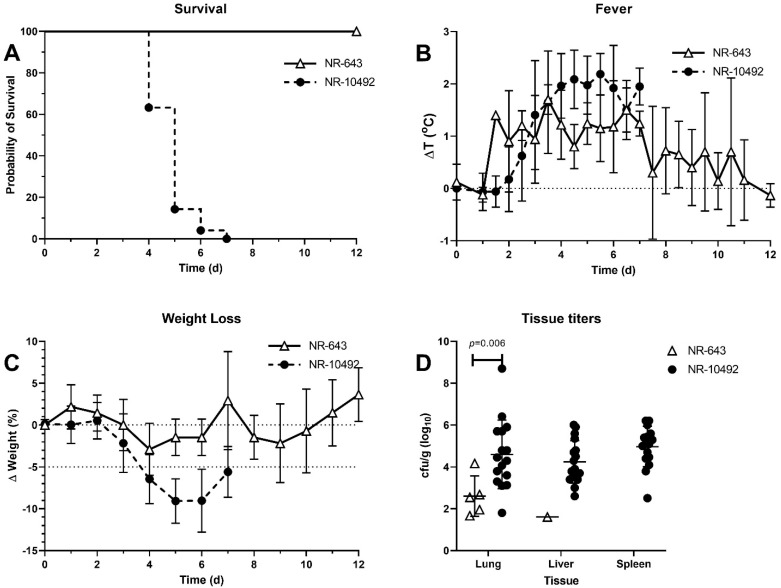
Virulence of NR-643 and NR-10492 in NZW rabbit model of pneumonic tularemia. NZW rabbits were exposed to 54 to 5523 CFU of NR-643 (n = 5) or 83 to 53,000 CFU of NR-10492 by aerosol inhalation. **A** total of 53 rabbits have been exposed to NR-10492, of which 30 were at a dose ≤5000 CFU. The graphs summarize studies performed over 10 years and include five rabbits infected with NR-643 and 53 infected with NR-10492. (**A**) survival out to 12 days after infection; (**B**) averaged twice daily body temperatures after infection plotted as change from baseline; (**C**) averaged daily change in weight after infection; (**D**) bacterial burden in lung, liver, and spleen at necropsy. Error bars show SD.

**Table 1 pathogens-10-00638-t001:** *F. tularensis* stocks used in this study.

*F. tularensis* Strain	Source	Catalog No.
SCHU S4	BEI Resources	NR-28534
SCHU S4	BEI Resources	NR-10492
SCHU S4	BEI Resources	NR-643
SCHU S4	Battelle Memorial Institute/The Ohio State University	FTS-635
MA00-2987	BEI Resources	NR-645

**Table 2 pathogens-10-00638-t002:** Information on the published reference SCHU S4 genomic DNA sequence and the substrains sequenced in this study.

Strain ID	BioSample Accession	GenBank Assembly Accession	Assembly Level
Reference (GenBank: AJ749949.2)	SAMEA3138185	GCA_000008985.1	Complete genome
NR-28534	SAMN02335346	GCA_000628925.1	Scaffold
NR-10492	SAMN02335347	GCA_000629005.1	Scaffold
NR-643	SAMN02335348	GCA_000628985.1	Scaffold
FTS-635	SAMN02335351	GCA_000628905.1	Contig
NR-645 (GenBank: CP012372.1)	SAMN02595231	GCA_001267475.1	Complete genome

**Table 3 pathogens-10-00638-t003:** Regions of difference between reference-assisted assembly and the reference.

Samples	Reference Location	Putative RD	Note
NR-28534 & NR-10492	354106 to 379789	Inversion	This inversion has also been observed in a virulent type A1 strain NE061598 [24]
NR-28534 & NR-10492	1767864	C → T	Intergenic SNP between FTT_1698c (formate dehydrogenase) and FTT_r08 (5S ribosomal RNA)
NR-643 & FTS-635	24609	A → G	Intergenic SNP between two hypothetical proteins (FTT_0025c and FTT_0026c)
NR-643 & FTS-635	427432	A → -	Single base deletion causes a frameshift in protein FTT_0415: glgC
NR-643 & FTS-635	541270	TTTATATAAGT → -	11 bp intergenic deletion between FTT_0517 and FTT_0518
NR-643 & FTS-635	694308	A → G	Nonsynonymous SNP causes an amino acid change (E->G) in a hypothetical protein (FTT_0676)
NR-643 & FTS-635	826816	A → G	Nonsynonymous SNP causes an amino acid change (D->G) in CapA membrane protein (FTT_0807)
NR-643 & FTS-635	1419877	C → T	Nonsynonymous SNP causes an amino acid change (P->S) in a 3-oxoacyl-ACP synthase (FTT_1373)
NR-643 & FTS-635	1423162	A → G	Nonsynonymous SNP causes an amino acid change (S->G) in a 3-oxoacyl-ACP synthase (FTT_1377)
NR-643 & FTS-635	1540424	- → A	Single-base intergenic insertion between a hypothetical protein (FTT_1486c) and dephospho-CoA kinase (FTT_1487)
NR-643 & FTS-635	1634580	G → A	Nonsynonymous SNP causes an amino acid change (T->I) in a hypothetical protein (FTT_1573c)

## Data Availability

The datasets generated for this study are available on request to the corresponding author.

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
