# Peer review of "Identification of an Attenuated Substrain of Francisella tularensis SCHU S4 by Phenotypic and Genotypic Analyses"

_pathogens, 2021, doi:10.3390/pathogens10060638_

Round 1
Reviewer 1 Report
Tularemia is an important zoonotic disease, particularly with regards to its biothreat potential. Ensuring the robustness and reproducibility of animal models is a scientific and ethical imperative for scientific research. The data presented in this paper provides a robust argument for the importance of characterising bacterial challenge strains used in animal challenge models. These data are of direct relevance to the evaluation and development of therapeutics for tularemia, and conceptually, to the wider research community.
Minor comments
- Line 3, tularensis misspelt in title.
- Can the authors provide any further details on how the different SCHU S4 strains were propagated or maintained, if known. For example were they subcultured in different media?
- Statistical analysis. Significant differences in bacterial burdens were analysed using unpaired t-tests. Since differences for multiple organs were analysed (eg Figure 2), a test that allows for multiple comparison errors should be applied (eg ANOVA or the Bonferroni’s method). It would be visually helpful if any significant difference is indicated on Figure 2C histogram.
- The rat challenge dose, as presented on Figure 1, was reported as average lung deposition from 3 representative rats. The methods also describe bacterial enumeration of aerosol impinger fluid to calculate the presented dose, although presented dose figures are not provided. How did the deposited dose differ from the presented dose and if differences were observed, was the magnitude of the difference different for the strains?
- Data is collated from a large number of rabbit infections with broad range of doses. Can the authors provide more details regarding the distribution of challenge doses for each strain?
Author Response
The authors would like to like thank the Reviewer for the constructive critique of our manuscript. We have provided additional experimental details relevant to the rat and infection studies and performed the statistical analysis recommended by the Reviewer after consultation with our biostatistician. Our point-by-point response to the reviewer’s comments are listed below:
- Line 3, tularensis misspelt in title.
The spelling of Francisella tularensis has been corrected and italicized
- Can the authors provide any further details on how the different SCHU S4 strains were propagated or maintained, if known. For example were they subcultured in different media?
These details are provided in the Materials & Methods section under section 4.3, line 353. There were differences between labs regarding propagation methods, but within each lab all cultures were prepared and propagated the same. So, all rat aerosol exposures regardless of F. tularensis stock used were propagated in the same manner and all rabbit aerosol exposures were propagated in the same manner regardless of F. tularensis stock used.
- Statistical analysis. Significant differences in bacterial burdens were analysed using unpaired t-tests. Since differences for multiple organs were analysed (eg Figure 2), a test that allows for multiple comparison errors should be applied (eg ANOVA or the Bonferroni’s method). It would be visually helpful if any significant difference is indicated on Figure 2C histogram.
The results were re-analyzed using repeated measures two-way ANOVA with Bonferroni’s multiple comparison test. The P values are summarized in Figure 2 panel C and the actual P values are reported in the caption.
- The rat challenge dose, as presented on Figure 1, was reported as average lung deposition from 3 representative rats. The methods also describe bacterial enumeration of aerosol impinger fluid to calculate the presented dose, although presented dose figures are not provided. How did the deposited dose differ from the presented dose and if differences were observed, was the magnitude of the difference different for the strains?
Using an approximate weight of 180 g for 16-week old female Fischer 344 rats to estimate the respiratory minute volume by the Guyton formula, the calculated presented doses ranged between 4 to 7-fold higher than the lung deposition. The additional detail was added to the Figure 1 caption (line 101).
|
Bacteria Stock |
Average Presented Challenge Dose (CFU/rat) |
Lung Deposition (CFU/rat) |
Ratio Presented Dose to Lung Deposition |
|
NR-643 |
285 |
50 |
6 |
|
NR-28543 |
669 |
95 |
7 |
|
NR-645 |
433 |
112 |
4 |
|
NR-10492 |
291 |
69 |
4 |
|
FTS-635 |
681 |
156 |
4 |
- Data is collated from a large number of rabbit infections with broad range of doses. Can the authors provide more details regarding the distribution of challenge doses for each strain?
We have added more detail in the text (lines 173-179) and to Figure 5 regarding the distribution of doses for the rabbits to NR-10492 and NR-643.
Reviewer 2 Report
This is a very important manuscript for the field. It examines the virulence of several “variants” of Francisella tularensis subspecies tularensis strain, SCHU S4, the archetypal virulent type A strain used for studying infection and immunity in a variety of animal models. In the present manuscript the authors show that at least one variant of SCHU S4, NR643, currently available in the pathogen collection managed by BEI Resources is far less virulent in a rat and rabbit aerosol model of primary infection than other variants obtained from the same source. Moreover, they show putative mutations within this strain that could account for its decreased virulence. Interestingly, this strain is not mutated in the FSC0918 or FSC0919 genes that has been shown to be the primary reason for the attenuation of a different variant of SCHU S4, FSC043. Indeed, NR643 appears to be intermediate in virulence between “true’ wild-type SCHU S4 and FSC043. Most importantly, this manuscript reiterates a truism for all infection and immunity studies, that is to ensure that they are performed with an appropriately virulent strain of the pathogen under investigation. Nowadays, this is the exception rather than the rule. In the case of NR643, others have shown it to be less virulent for mice than other strains of F. tularensis subsp. tularensis, and have wrongly concluded that SCHU S4 is less virulent than comparator strains. This manuscript sets the record straight for this SCHU S4 variant going forward. I agree with the authors that strain NR643 should not be considered to be a bona fide SCHU S4 variant with respect to virulence and should be avoided for all studies where a fully virulent variant of the pathogen is required to obtain a meaningful result.
I do have a few minor suggestions for the authors which require minimum effort, but I believe will result in an improved manuscript. These are as follows:
Line 42. Should clarify that mortality rate is in the absence of effective therapy.
Figure 1. Authors should include the range as well as the average dose.
Figure 3. This figure would be improved if selected arrowed sites could be shown magnified within a separate box within each panel.
Figure 5. Black / white and open or closed symbols and lines are easier on the eye than gray.
Line 281. Should also include reference 28.
Line 436. Can the authors confirm that the genomic sequencing was performed on the minimally-passaged bacteria and not the bacteria that were passaged on multiple media for animal challenges which could have contributed to the observed mutations?
General comment. There is a known variant of SCHU S4 called SCHU S5 that is streptomycin-resistant and the latter somewhat less virulent than the former in mice. It might be interesting to quickly determine whether or not strain NR643 is also resistant to this antibiotic.
Author Response
The authors would like to like thank the Reviewer for the constructive critique of our manuscript and for endorsing the use of a truly virulent challenge strain for critical studies. We have revised the text and figures to address most of the Reviewer’s comments and provided additional responses to Reviewer’s suggestions to further improve the manuscript. Our point-by-point response to the reviewer’s comments are listed below:
Line 42. Should clarify that mortality rate is in the absence of effective therapy.
Added the additional text as requested.
Figure 1. Authors should include the range as well as the average dose.
Revised the figure legends to include the dose range. We also included the dose range in the caption for Figure 2 and in Figure 4
Figure 3. This figure would be improved if selected arrowed sites could be shown magnified within a separate box within each panel.
Figure 3 was intended to show the relative size and distribution of the lesions, not the character of the lesion. This necessitated the use of a low magnification image. We can't magnify the current image without losing resolution. The description in the text references the pathogenesis study done in rats if the reader would like to see the progression of the lesions.
Figure 5. Black / white and open or closed symbols and lines are easier on the eye than gray.
Revised as requested
Line 281. Should also include reference 28.
Added Lindgren et al. (ref 28) as requested.
Line 436. Can the authors confirm that the genomic sequencing was performed on the minimally-passaged bacteria and not the bacteria that were passaged on multiple media for animal challenges which could have contributed to the observed mutations?
The stocks were not passaged more than once prior to sequence. This detail has been added to Section 4.6. Genome sequencing.
General comment. There is a known variant of SCHU S4 called SCHU S5 that is streptomycin-resistant and the latter somewhat less virulent than the former in mice. It might be interesting to quickly determine whether or not strain NR643 is also resistant to this antibiotic.
While we agree that SCHU S5 may be of some interest in understanding virulence and antibiotic resistance of F. tularensis, it is beyond the scope of this article which is focused on the observed differences in virulence of the SCHU S4 stocks.